# Analysis of Postural Stability Following the Application of Myofascial Release Techniques for Low Back Pain—A Randomized-Controlled Trial

**DOI:** 10.3390/ijerph20032198

**Published:** 2023-01-26

**Authors:** Piotr Ożóg, Magdalena Weber-Rajek, Agnieszka Radzimińska, Aleksander Goch

**Affiliations:** Department of Physiotherapy, Collegium Medicum in Bydgoszcz, Nicolaus Copernicus University in Toruń, 85-067 Bydgoszcz, Poland

**Keywords:** postural stability, low back pain, myofascial release

## Abstract

Introduction: Low back pain (LBP) is one of the most frequently observed disorders of the musculoskeletal system in the modern population. It is suggested that myofascial disorders in the highly innervated thoracolumbar fascia (TLF), reported in patients with LBP, may be an underlying cause of the ailment. Research also confirms that patients with LBP demonstrate poorer postural stability compared with individuals without the condition. Myofascial release techniques (MFR) are additional therapeutic options that complement existing therapies and help provide a more holistic treatment for chronic LBP (CLBP). Objective: Evaluation of changes in postural stability following one MFR intervention applied to CLBP subjects immediately after manual therapy and after a month. It was hypothesized that postural stability is going to aggravate immediately after the MFR intervention and improve one month after treatment compared with the baseline results before the treatment. Methods: 113 patients with CLBP participated in a randomized-controlled trial. The experimental group (n = 59) received one MFR intervention, whereas the control group (n = 54) did not receive any therapeutic intervention. Posturography was performed to determine experimental group’s immediate response to the therapy and to evaluate the experimental and control groups’ responses to the therapy one month after the intervention. Results: Only 2 out of 12 comparisons of stabilometric parameters demonstrated reliable effects that are in line with our research hypotheses. Even though both comparisons were observed for therapy outcomes within the experimental group, no reliable differences between the groups were found. Conclusions: A single MFR treatment in the TLF did not affect postural stability in CLBP patients in the experimental group. Further studies are needed to extend the findings by performing a series of holistic MFR treatments applied to a larger area of the body surface that would induce more general tissue changes and thus having a greater impact on postural stability.

## 1. Introduction

Low back pain (LBP) is one of the most frequently observed disorders of the musculoskeletal system in the modern population. Low back pain is considered chronic (CLBP) when lasting over 3 months or persistent even after healing of damaged tissues. CLBP occurrence increases with age, reaching its peak in the 5th and 6th decades of life. People suffering from CLBP may experience pain even in response to a small stimulus, and the long-term discomfort may increase the level of depression and odds of sleep disorders, thus significantly deteriorating the quality of life [1,2]. 

Numerous authors agree that a modest percentage of pain ailments are caused by specific serious causes, whereas 90% of cases are non-specific low back pain without an established cause [3,4,5,6,7]. It is believed that myofascial disorders in the highly innervated thoracolumbar fascia (TLF) may be among the causes of non-specific low back pain. Patients suffering from LBP have significantly more active trigger points in the erector spinae as well as the quadratus lumborum [8,9], which may contribute to muscle stiffness, limited mobility, and even muscle weakness [10,11,12]. Moreover, Langevin et al. and Ranger et al. proved in their studies that there is a significant shortening and thickening of the TLF area in individuals with LBP [13,14]. 

Individuals suffering from LBP are prone to problems with postural stability. Most authors confirm that average values of center of pressure (COP) velocity obtained in tests conducted in a free-standing position are significantly higher in adults with CLBP compared with healthy individuals [15,16,17]. The same relationship is observed in comparisons of COP sway areas (SA), defined by the authors as the elliptical area containing 90% of the COP position during the given test time [15,17,18]. Patients with LBP exhibit higher SA values compared with healthy population. Furthermore, the authors emphasize that there are significantly higher values for anterior–posterior and lateral COP displacement among individuals with LBP, and these are present among majority of adults aged 30 to 60 [19,20,21]. 

Myofascial release techniques (MFR) are additional therapeutic options that, according to current recommendations, may complement manual joint mobilization techniques, physical exercises and pain neuroscience education and help provide a more holistic treatment for CLBP [22]. Manual therapies that focus on the myofascial system, which can be combined with the patient’s active movement, are aimed at improving flexibility and gliding between layers of soft tissues. Stimulation of mechanoreceptors located in the connective tissue may mechanically improve proprioception, reduce the excessive activity of the paraspinal muscles and sympathetic activity of the autonomic nervous system, thus adding significant clinical value to the therapy [23,24]. MFR techniques requiring the maintenance of long-term compression, e.g., in the case of myofascial trigger points therapy, can be performed with the use of specially designed ergonomic supporting tools (instrumental therapy) that facilitate the maintenance of an even compression and improve the comfort of performing the technique [25].

The reliability of the immediate impact of MFR on patients with LBP was confirmed several times by authors who used objective research tools. The findings in those studies confirm that even a single, isolated MFR intervention significantly improves the sliding movements between TLF layers [23], increases lumbar spine range of motion [26], decreases resting activity of the erector spinae and multifidus muscles [27] and, according to Shah et al., improves lumbar paraspinal blood flow [28]. The use of MFR therapy may also affect postural stability. According to the Manheim theory, intensive stimulation of connective tissue receptors leading to a reflex change in muscle tone and the initiation of local micro-inflammation may lead to short-term and transient postural stability disorders immediately after the intervention [29]. Nevertheless, the significant reduction in pain intensity and improvement in functional performance, which are observed after the use of MFR therapy in people with CLBP [23,30,31], should lead to improved postural stability in the long-term evaluation. 

To our knowledge, an analysis of the immediate postural stability changes following one isolated MFR session has not been presented. The aim of this study was to evaluate if a single session of MFR performed in a group of CLBP patients led to any changes in the postural stability immediately after the session, as well as one month after the intervention. 

## 2. Methods

### 2.1. Study Design

The study was conducted at the Department of Physiotherapy Collegium Medicum in Bydgoszcz, Nicolaus Copernicus University in Toruń and at the OsteoClinic-Osteopathy and Physiotherapy Clinic in Toruń. The sample size was calculated by the formula for the minimum sample size (*N_min_*) [32]: $$N_{min}=\frac{P\left(1-P\right)}{\frac{e^{2}}{z^{2}}+\frac{P\left(1-P\right)}{N}}$$

The general population (*N* = 169) consisted of people who came to the OsteoClinic due to LBP in 2018, among whom the qualification process for the study was carried out. Based on the assumptions made (sample proportion = 70 % (*P* = 0.7), confidence level = 95 % (*z* = 1.96) and margin of error = 5% (*e* = 0.05)), it was established that a minimum of 111 people should complete the study. The study was carried out from June 2019 to March 2020. A total of 158 adults participated in the process of qualifying for the study. For participant enrollment the researchers used an interview questionnaire (including basic data, such as gender, age, height and weight, as well as questions about the duration of ailments, sports activity, type of work and the most common position during its performance, injuries and surgeries, as well as the presence of comorbidities and contraindications to the therapy conducted in the study) and functional assessment (neurological examination) performed according to the protocol set out below.

Study inclusion criteria were as follows: (1) aged 40 to 60; (2) CLBP for more than 3 months. Study exclusion criteria were as follows: (1) having any neurological symptoms (tested via a neurological examination including: the slump test, the ankle clonus test and the Babinski reflex test, tension tests for the sciatic nerve (bowstring test and straight leg raise) and the femoral nerve (in the prone and side-lying position), assessment of ankle and knee reflexes, evaluation of the muscle strength and superficial sensation); (2) positive result in pain provocation tests: spinal axial compression and maximum compression of the intervertebral foramen in the area of the lumbosacral spine; (3) the presence of contraindications for MFR therapy (active malignant disease, deep vein thrombosis, aneurysms, infectious diseases, viral and bacterial infections, acute inflammation and fever); (4) history of a spine injury or received surgical treatment; (5) use of physiotherapy during the last 6 months; (6) coexisting conditions such as osteoporosis, diabetes, cancer and pregnancy or cardiovascular system, rheumatic, psychic, digestive system and gynecological diseases.

Participants who qualified for the study were randomly assigned to two research groups using the simple randomization method [33]. Researchers prepared 119 sealed allocation envelopes, each containing a sheet of paper with an even or odd number (1–119). Each enrolled subject drew a sealed envelope. Participants who drew odd numbers (n = 60) were assigned to the experimental group (EG) and received MFR treatment, whereas those who drew even numbers (n = 59) were allocated to the control group (CG), in which no therapy was applied. The experimental group was evaluated three times: before the therapy, immediately after the therapy, and one month following the treatment. We assessed the control group twice: once before the therapy and then one month later. There was no blinding in the study. Therapy and examination in all cases was performed by one researcher (PO). However, the randomization process was carried out by another researcher (MWR). The subjects were asked to avoid any physiotherapy during the month between the MFR session and the control visit. 

At the evaluation visit one month after the intervention, one participant was excluded from the EG due to an ankle injury. Five participants were excluded from the CG (three subjects received physiotherapy within the last month and two subjects did not come to the control visit). Overall, 113 participants completed the study. The flow of all participants through the phases of the study is shown in Figure 1. The EG comprised 27 women and 33 men, aged 41–60 (Me = 49.36; SD = 5.91), whereas the CG comprised 28 women and 26 men, aged 41–60 (Me = 48.91; SD = 5.38). The initial degree of pain intensity according to the VAS scale was similar in both groups and was 5.08 (Me = 5; SD = 1.59) in the EG and 5.22 (Me = 5; SD = 1.72) in the CG. The majority of the study participants claimed to be performing mainly white-collar work (EG = 72.27%; CG = 53.70%) in the sedentary position (EG = 77.97 %; CG = 53.70%). 

### 2.2. Measurements 

Postural stability was evaluated using a posturography test that registers and analyzes COP movement. The researchers registered and assessed stabilometric parameters using a Loran Engineering E.P.S./R1 stabilometric platform equipped with 2304 resistive sensors and Biomech Studio 2013 software. The assessment of postural stability was carried out in static conditions on a stable surface and in a free-standing position. Two tests were conducted—with eyes open (EO) and with eyes closed (EC). Each test was conducted in accordance with recommendations in the available literature and lasted 30 seconds [35]. Study participants were instructed to remain silent and stand freely while avoiding any movement of the body. Moreover, study participants could not see COP displacements on the screen, because, according to the observations of Wulf et al., watching COP displacements may lead to improved measurement results [36]. During the test with eyes opened the subjects were asked to focus their sight on a point at eye level at a distance of two meters. The participants stood on the platform on both feet, barefoot, with feet equally leveled to the hips and arms free alongside the body (Figure 2). 

The results of the following posturographic parameters characterized by excellent reliability (with a correlation coefficient > 0.9) [37] were analyzed: (1) COP distance [mm] (value of the total COP path length obtained during the study); (2) COP sway area [mm^2^] (COP-SA; defined in Biomech Studio as an elliptic area covering 90% of COP positions); (3) COP sway velocity [mm/s]. 

### 2.3. Intervention

One session of manual therapy involving MFR techniques was performed in the EG. The manual therapist used natural fascial release wax to ensure optimal tissue control and the highest quality of techniques. The techniques involved all three layers of TLF and were applied according to Luchau’s guidelines [38]. By performing techniques in the sequence described by Myers and Manheim, the researchers first engaged superficial tissue and then gradually moved onto deeper structures [29,39]. In the prone lying position, the following techniques were used: skin rolling [39], local stretching of erector spinae muscle using the cross-handed technique [29] and spinal and pelvic distraction [40]. In the side lying position: longitudinal stretching of the erector spinae and the multifidus muscles (fetal position) [40], the pin and stretch technique for the middle and posterior TLF layers [41] and the release of the quadratus lumborum (direct stretch technique [40] and pin and stretch [41]). All techniques were used on both sides of the body. The complete manual therapy procedure lasted 40 minutes in each case.

According to the recommendations of Manheim and Riggs, the strength of touch was adjusted to the individual subjective feelings of the patient during the procedure. Each of the patients was informed that the procedure should not be painful, and the fully acceptable sensations are “stretching”, “tightening” or “burning” of the treated tissue associated with the feeling of its relaxation [29,40]. In addition, the patients’ attention was drawn to the essence of the possibility of continuous, calm breathing and the lack of the need for active defensive contraction of the treated tissues. In the case of negative feelings that went beyond the above instruction, the patient immediately informed the therapist, who reduced the strength of the technique.

### 2.4. Statistical Analyses

R statistical environment (ver. 3.6.2, R Foundation for Statistical Computing, Vienna, Austria), was used for data operations, reported data analyses and visualizations. The data.table library was used for carrying out operations on data. Elementary R functions, ggplot2 and bayesplot libraries were used for plotting graphs [42]. The researchers tested the research hypotheses by means of hierarchical Bayesian regression employed using the brms library [43]. Matching the data to the null model allowed for the therapeutic effects in the EG to be analyzed. However, it is essential to point out that the null model presented the patient only with a random effect. The next step engaged determining parameters of the 0/1 measurement model for coding therapy effects (immediately and one month after the intervention vs. before intervention). The measurement model’s greater predictive power compared with the null model suggests the validity of the hypothesis regarding the effectiveness of therapy in the EG. That is, of course, as long as the observed effects are in the assumed direction. We performed a stepwise comparative analysis of the experimental group and the control group. The null model was fitted with data first, followed by the main effects model (EG model) and the main effects and interaction model (the EGI model). The EG model comprised main effects for the group (coded 0 for the control group and 1 for the experimental group) and the measurement (coded 0 for pre-intervention and 1 for one month following the therapy). The main and interaction effects between a group and the therapy were included in the EGI model. The EGI model provided the best fit, implying that a change in the control group parameter is reliably different from a change in the experimental group parameter.

By employing two information criteria established with the cross-validation method—the leave-one-out information criterion (LOOIC) and the k-fold information criterion (KFOLDIC)—researchers compared predictive power of the estimated regression models. Both statistics are used to measure a model’s predictive power outside the sample, i.e., the model’s ability to accurately forecast new observations. Because the LOOIC and KFOLDIC values are based on logarithms of probability, discrepancies in their values always imply compelling evidence in favor of the better model. This can be explained by the fact that discrepancies between 110 and 100 and between 1010 and 1000 are considered equally compelling evidence—10 units, regardless of the baseline statistics—in favor of the better-fitted model. The LOOIC was the default statistic. Nevertheless, if the LOOIC approximation was unreliable, KFOLDIC was applied [44]. Lower values for both statistics indicate a better fit to the data. As with any information criterion, differences over 3 (absolute) units are weak evidence in favor of a better model. Differences of 4–7 represent medium-strong evidence, and differences greater than 10 represent compelling evidence in favor of the better model [45]. In addition, R-squared for Bayesian models using a continuous dependent variable was reported to provide an intuitive measure of the effect’s magnitude [46].

## 3. Results 

### 3.1. COP Distance 

The value for the COP distance (mm), was shown to be significantly higher than the average, thus yielding a strong positive skewedness (Figure 3). 

In comparisons made in the EG, the measurement model fits the data significantly better than the null model, which is confirmed by the LOOIC difference of -44. Furthermore, the effect of time measurement accounted for 7% of the variance of the EO variable and for merely 1% of the variance of the EC variable (Table 1). These results suggest a lack of effect on the EC dimension. In our study, an increase in the average COP distance value for the EO dimension was observed immediately after the intervention; however, no reliable difference was found. On the contrary, we determined a reliably lower average COP distance value recorded one month following the intervention, compared with the result obtained before therapy (Figure 4). 

In comparisons of results between groups the model with the interaction effect did not fit the data better than the main effects model, which is confirmed by the KFOLDIC difference of 2, and the model did not differ from the null model (the KFOLDIC difference = −1) (Table 2). These results indicate lack of reliable differences for the COP distance variable between the groups (Figure 5), which is also confirmed by low values of the variance explained for interaction effects presented in Table 2. 

### 3.2. COP Sway Area

The value of COP-SA is highly skewed and might follow a lognormal distribution, similarly to the COP distance variable (Figure 6). 

In comparisons made in the EG, the measurement model fits the data slightly better than the null model, which is confirmed by the LOOIC difference of -5. Furthermore, the effect of time measurement accounted for 1% of the variance of the EO dimension and for 0% of the variance of the EC variable (Table 1). These results suggest a lack of effect on both dimensions of the COP-SA variable (Figure 7). 

In comparisons of results between groups the model with the interaction effect did not fit the data better than the main effects model, which is confirmed by the KFOLDIC difference of 3, nor did the null model, which is confirmed by the KFOLDIC difference of 3. These results indicate a lack of reliable differences for the COP-SA variable between the groups (Figure 8), which is also confirmed by the quite low values of the variance explained for interaction effects in Table 2. 

### 3.3. COP Sway Velocity

Similarly to COP distance and COP-SA, COP sway velocity is characterized by a noticeable positive skewedness, and lognormal distribution was also used for the analysis of COP sway velocity (Figure 9). 

In comparisons made in the EG, the measurement model fits the data better than the null model, which is confirmed by the LOOIC difference of −9. Furthermore, the effect of time measurement accounted for 7% of the variance of the EO dimension and for merely 1% of the variance of the EC variable (Table 1). These results suggest a lack of effect on the EC dimension. The COP sway velocity value for the EO dimension was reliably higher both immediately after the intervention and one month following the treatment than before the intervention (Figure 10). 

In comparisons of results between groups, the model with the interaction effect fits better the data better than the main effects model, which is confirmed by the LOOIC difference of −7, or the null model, which is confirmed by the LOOIC difference of −3. Main effects, interaction time and groups accounted for 6% of EO variance and 2% of EC variance (Table 1). We observed a difference in COP sway velocity values in the study groups before the intervention—the COP sway velocity was greater in the CG than in the EG (Figure 11); however, the difference vanished one month after the intervention. 

## 4. Discussion

The obtained results of the stabilometric parameters presented in this study do not allow us to conclude whether there is any credible impact of a single MFR procedure on postural stability in CLBP patients. Based on theoretical assumptions about the possible impact of MFR interventions on postural stability presented in the Introduction section of this manuscript [29,30,31], it was hypothesized that postural stability was going to aggravate immediately after the MFR intervention and improve one month after treatment compared with the baseline results before the treatment. Therefore, the values of all posturographic parameters should reliably increase immediately after the applied MFR intervention, and after one month, show reliably lower values compared with the results before the therapy. Moreover, of course the resulting changes in parameter values should be greater in the EG than in the CG. Only 2 out of 12 comparisons of stabilometric parameters demonstrated reliable effects that were in line with our research hypotheses. Even though both comparisons were observed for therapy outcomes within the EG, no reliable differences between the groups were found. The first part of the hypothesis, which was based on the Manheim’s assumptions presented in the Introduction, assumed that postural stability is going to aggravate immediately after the MFR procedure [29]. There are mechanisms that can explain the correctness of the thesis proposed by Manheim. During MFR intervention, the receptors of the proprioceptive system located in richly innervated connective tissue are mechanically stimulated, which is a significant part of the posture control system, playing a major role in maintaining a standing body posture [15]. Intensive stimulation of mechanoreceptors may affect the reflex change of muscle tone, which, especially in the area of the postural muscles, may temporarily interfere with the control of postural stability. Moreover, the MFR procedure leads to the initiation of local microinflammation, stimulating and controlling the process of the remodeling of the collagen fibers of the connective tissue. Inflammation may disrupt the afferent information transmitted from proprioreceptors to the central nervous system, temporarily hindering the control of postural stability [23]. Results measured in the EG reliably confirmed this correlation only for COP sway velocity in the eye-open test. Nevertheless, the results of the comparisons of the remaining stabilometric parameters clearly did not confirm the relationship suggested by Manheim. Both the COP sway area and the total COP path length did not change significantly immediately after the treatment; therefore, it is not possible to conclude whether the obtained results support the above-mentioned part of the hypothesis. Moreover, the author had not specified which body area was treated. It is essential to note that subjecting certain areas to manual therapy may contribute to the onset of postural stability disorders. A lack of reliable results following a single procedure is in accordance with the theoretical assumptions by Manheim [29] and Earls and Myers [41], according to whom achieving general changes in body posture and postural stability may require a series of holistic MFR treatments applied to a larger area of the body. The lack of deterioration in postural stability immediately after the intervention, as well as its improvement after a month compared with the initial state, may be due to the fact that the LBP treatment was applied only locally to the TLF area. On the contrary, treatment applied to another body part, e.g., lower limbs or the whole body, could lead to obtaining results consistent with the research hypotheses presented in this study.

According to some authors, Postural stability disorders in patients with LBP is caused by alteration of reflexive muscle activation in the trunk and lower limbs, which are the effector organs of the posture control system [47,48]. It is assumed that pain stimulation in the lower spine leads to reflex muscle contraction that is meant to limit mobility and stabilize lower body segments, thus protecting damaged structures and, in people with root pain, centralize ailments. Prolonged increased tension and abnormal muscle coordination may cause their fatigue and, consequently, lead to problems with postural stability, at the same time aggravating existing ailments [49]. Moreover, Nijs et al. emphasize the importance of the reactions of the autonomic nervous system to chronic nociceptive stimulation in the form of increased sympathetic activation. According to this interpretation, myofascial changes and postural stability disorders in people with CLBP may be consequences of long-term pain rather than causes of the pain [50]. Thanks to the intensive development of training, MFR is gaining more popularity among physiotherapists and manual therapists. However, one should bear in mind the history of manual therapy, according to which the basics of the techniques used today, focused on the myofascial system, date back several dozen years. Today, thanks to scientific research using more and more advanced apparatus, we can assess whether the use of a natural form of influence such as touch can objectively improve the health and functional fitness of patients with LBP. It is believed that the noticeable improvement in the functioning of the fascial system after the use of MFR results from increased tissue hydration and improved drainage of inflammatory mediators and metabolic waste, which may reduce chemical irritation of the autonomic nervous system nerve endings and nociceptive somatic endings [51]. According to Shah et al., the use of even a single MFR treatment improves blood circulation around the paraspinal muscles of the lumbosacral spine [28]. From the biomechanical point of view, the improvement of the sliding between the individual layers of the TLF, which is noticeable in imaging studies, may turn out to be significant [23], which according to Stecco et al. is crucial for the efficient functioning of the locomotor system [52]. The use of MFR therapy, which is meant to reduce muscle activity [27] and improve proprioception, could also affect postural stability among people with LBP. 

To our knowledge, an analysis of immediate changes in postural stability following one isolated MFR intervention has not been presented, which indicated the validity of our objective assessment of these parameters using a posturography test. The literature reports available so far in this area present only an assessment of postural stability in people with LBP using simple functional tests such as the Y-balance test, used to evaluate dynamic stability while standing on one foot, [53] and the Supine Bridge Test, used to evaluate core stability particularly in the low back and hip regions [54]. Nevertheless, in both studies, the MFR was not an isolated treatment but a supplemental therapy to core stability exercises. Even though the study by Mavajian et al. showed a significant improvement in dynamic stability following a single MFR session (in the TLF area) with core stability exercises, their research was conducted only on a small pilot group of people with CLBP (n = 10) and without a control group [53]. Furthermore, Ozsoy et al. did not use MFR therapy performed by a qualified manual therapist, and patients performed self-massage using a special roller three times a week over a period of six weeks, thus complementing their core stability exercise program. The combination of two forms of therapy allowed achievement of a greater progress in core stability endurance compared with the control group who just performed core stability exercises [54].

The results of this study did not confirm the assumptions presented in the research hypotheses. Nevertheless, some of the results obtained may have clinical application for practitioners working with patients with CLBP. Due to the fact that in this study a single MFR session did not deteriorate postural stability, it seems that this type of local intervention can be used without fear of intensifying the postural stability disorders associated with CLBP. 

## 5. Study Limitations

We are aware of the limitations of our study. These include the lack of use of sham therapy in the control group, the lack of blinding, different arrangements in outcome assessment between the two groups and the short intervention period. The was no pain assessment and no comparison of demographic data or patient characteristics. Additionally, there was limited control over study participants, especially in their daily activities and psycho-emotional aspects in the period between measurements, which could have impacted postural stability results.

## 6. Conclusions

A single MFR treatment in the TLF did not aggravate postural stability immediately after the therapy in CLBP patients in the experimental group. Moreover, after one month, postural stability did not improve compared with the results recorded before the treatment. The values of the stabilometric parameters one month after the intervention did not change significantly in the experimental group compared with the control group.

Further studies are needed to extend our findings by performing a series of holistic MFR treatments applied to a larger area of the body surface that could induce more general tissue changes, and thus having a greater impact on postural stability. 

## Figures and Tables

**Figure 1 ijerph-20-02198-f001:**
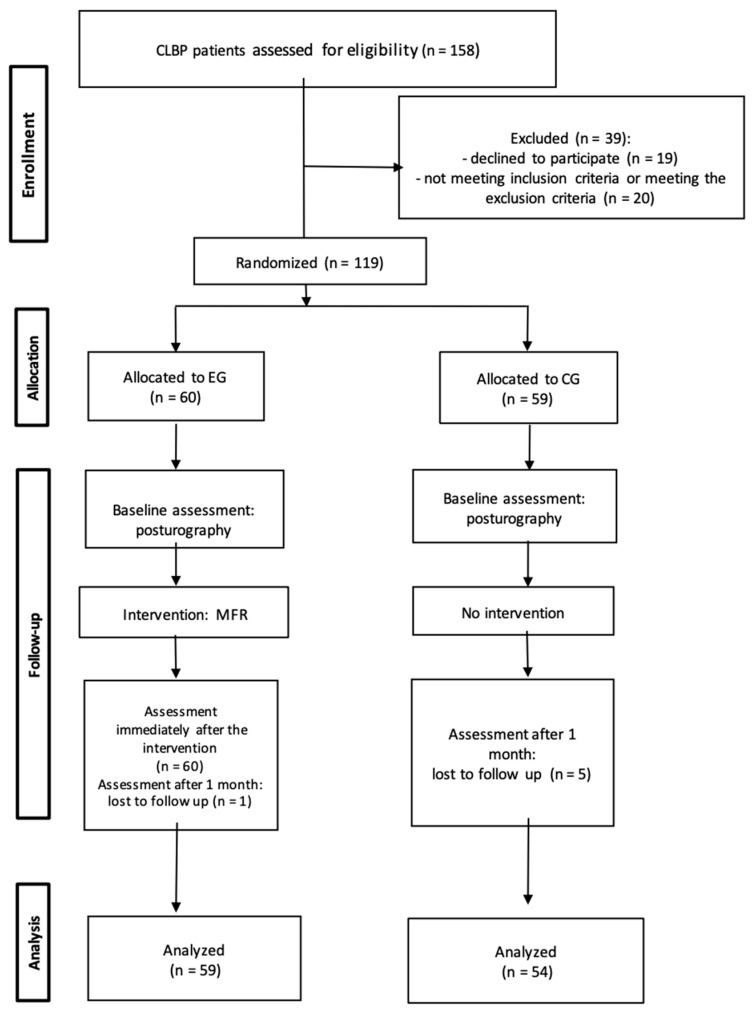
Study flowchart [34]. CLBP—chronic low back pain; EG—experimental group; CG—control group; MFR—myofascial release.

**Figure 2 ijerph-20-02198-f002:**
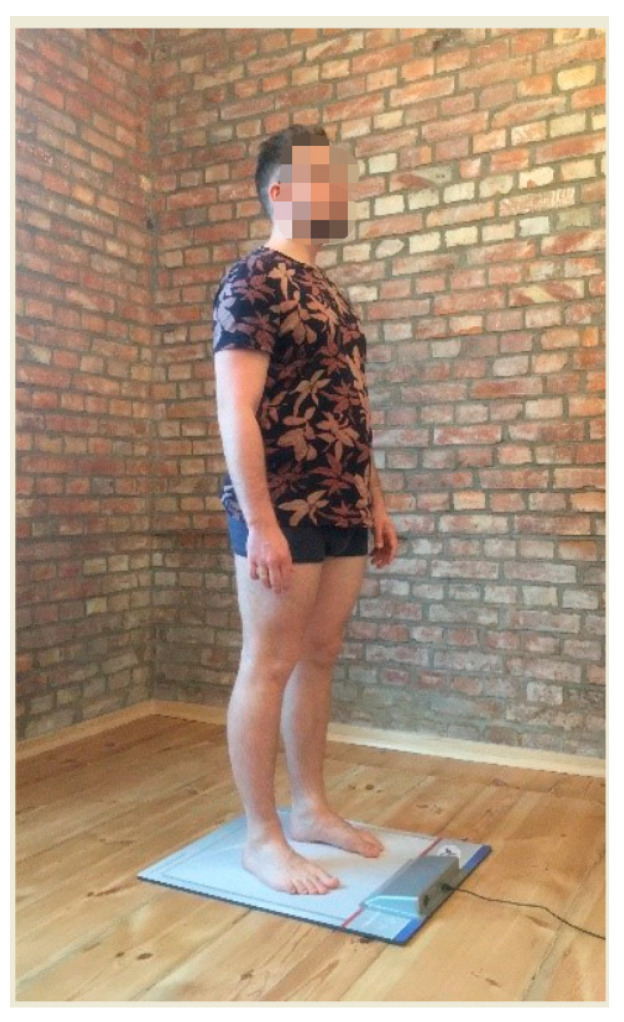
Evaluation using a stabilometric platform. Author’s content.

**Figure 3 ijerph-20-02198-f003:**
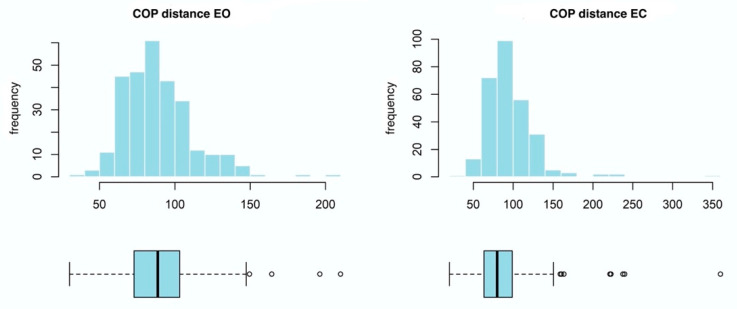
Box plots and histograms for the two-dimensional COP distance variable. The graphs were constructed from all the data gathered from both groups. COP—center of pressure; EO—eyes open; EC—eyes closed.

**Figure 4 ijerph-20-02198-f004:**
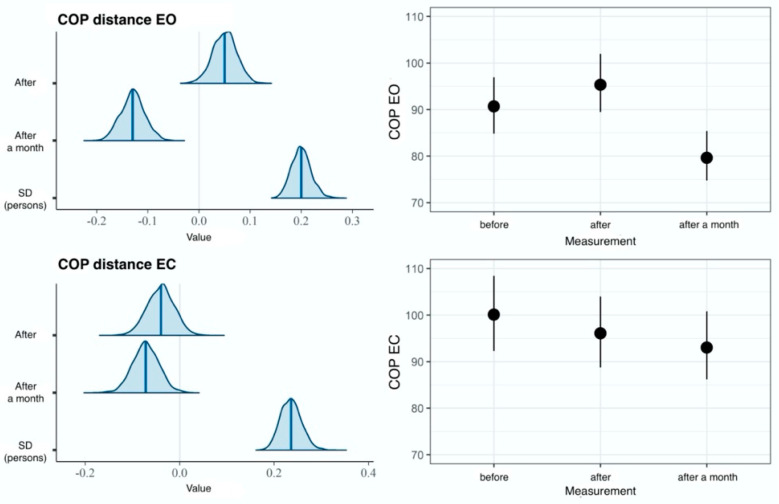
Results of hierarchical two-dimensional regression analysis: the dependent variable is the COP distance, and the main effect is the measurement in the experimental group. Left panels: a posteriori distributions of regression coefficients. Shaded areas denote 95% confidence level. Right panels: estimated marginal means. The points denote a posteriori distribution mean values, and vertical lines represent 95% confidence intervals. COP—center of pressure; EO—eyes open; EC—eyes closed.

**Figure 5 ijerph-20-02198-f005:**
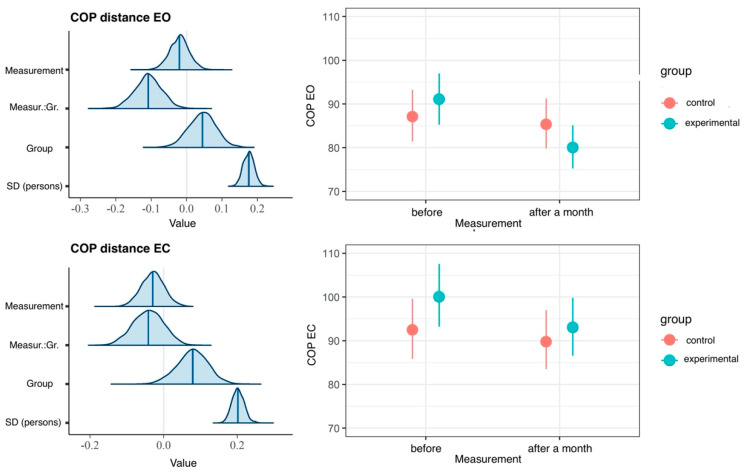
Results of hierarchical two-dimensional regression analysis: the dependent variable is the COP distance, whereas the predictors are the dependent variable and the group. Left panels: a posteriori distributions of regression coefficients. Shaded areas denote 95% confidence level. Right panels: estimated marginal means. The points denote a posteriori distribution mean values, and vertical lines represent 95% confidence intervals. COP—center of pressure; EO—eyes open; EC—eyes closed; Measur.:Gr.—Measurement: Group.

**Figure 6 ijerph-20-02198-f006:**
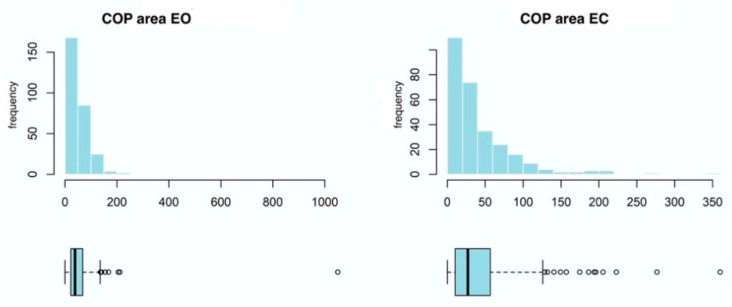
Box plots and histograms for the two-dimensional COP-SA variable. The graphs were constructed from all the data gathered from both groups. COP—center of pressure; EO—eyes open; EC—eyes closed.

**Figure 7 ijerph-20-02198-f007:**
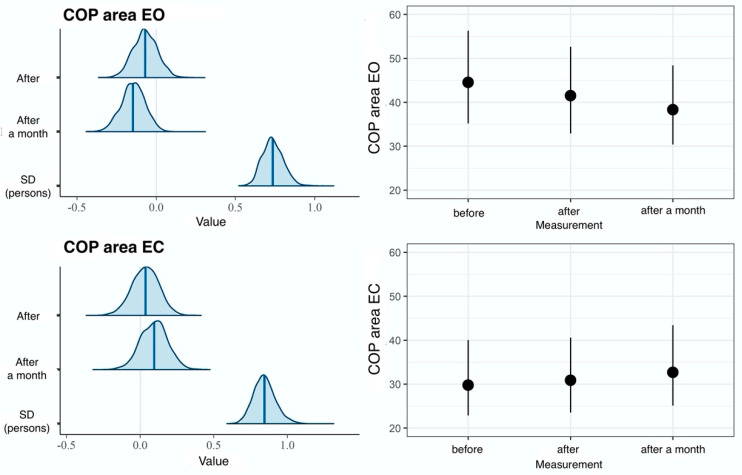
Results of hierarchical two-dimensional regression analysis: the dependent variable is the COP-SA, and the main effect is the measurement in the experimental group. Left panels: a posteriori distributions of regression coefficients. Shaded areas denote 95% confidence level. Right panels: estimated marginal means. The points denote a posteriori distribution mean values, and vertical lines represent 95% confidence intervals. COP—center of pressure; EO—eyes open; EC—eyes closed.

**Figure 8 ijerph-20-02198-f008:**
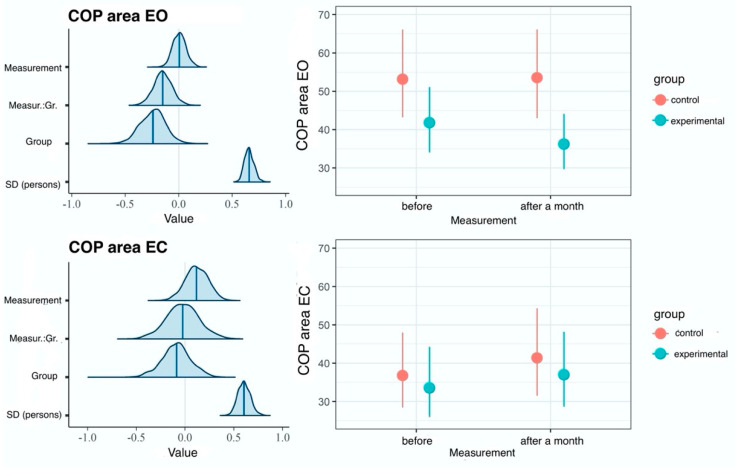
Results of hierarchical two-dimensional regression analysis: the dependent variable is the COP-SA, whereas the predictors are the dependent variable and the group. Left panels: a posteriori distributions of regression coefficients. Shaded areas denote 95% confidence level. Right panels: estimated marginal means. The points denote a posteriori distribution mean values, and vertical lines represent 95% confidence intervals. COP—center of pressure; EO—eyes open; EC—eyes closed; Measur.:Gr.—Measurement: Group.

**Figure 9 ijerph-20-02198-f009:**
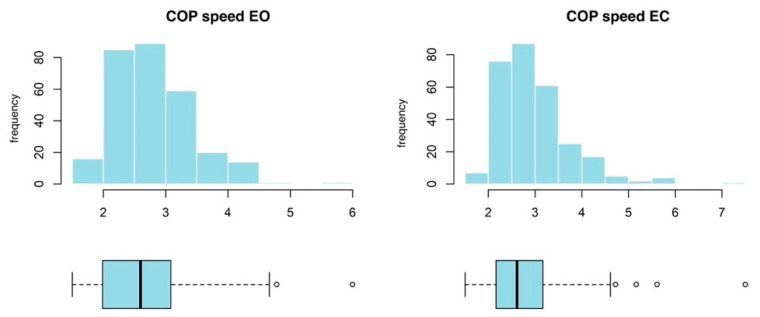
Box plots and histograms for the two-dimensional COP sway velocity variable. The graphs were constructed from all the data gathered from both groups. COP—center of pressure; EO—eyes open; EC—eyes closed.

**Figure 10 ijerph-20-02198-f010:**
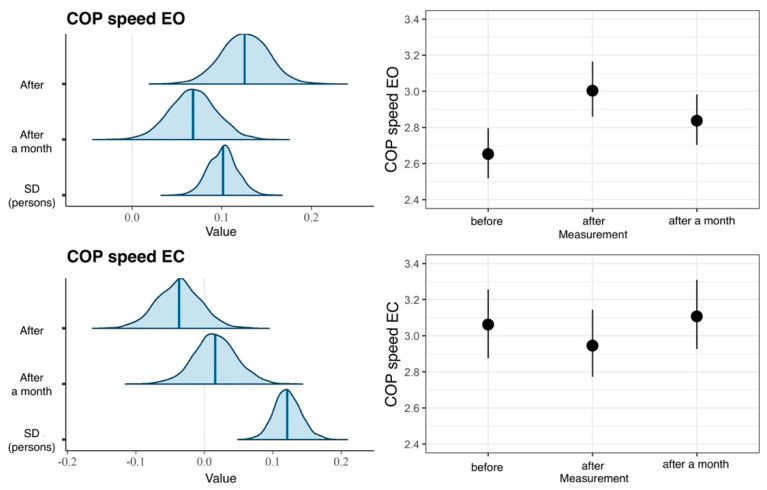
Results of hierarchical two-dimensional regression analysis: the dependent variable is the COP sway velocity, and the main effect is the measurement in the experimental group. Left panels: a posteriori distributions of regression coefficients. Shaded areas denote 95% confidence level. Right panels: estimated marginal means. The points denote a posteriori distribution mean values, and vertical lines represent 95% confidence intervals. COP—center of pressure; EO—eyes open; EC—eyes closed.

**Figure 11 ijerph-20-02198-f011:**
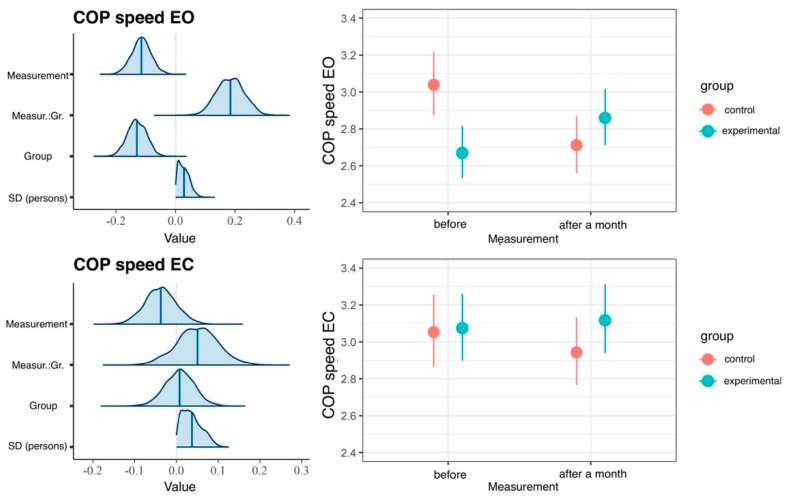
Results of hierarchical two-dimensional regression analysis: the dependent variable is the COP sway velocity, whereas the predictors are the dependent variable and the group. Left panels: a posteriori distributions of regression coefficients. Shaded areas denote 95% confidence level. Right panels: estimated marginal means. The points denote a posteriori distribution mean values, and vertical lines represent 95% confidence intervals. COP—center of pressure; EO—eyes open; EC – eyes closed; Measur.:Gr.—Measurement: Group.

**Table 1 ijerph-20-02198-t001:** Fitness of the regression models to the dataset used in this paper for testing changes in experimental group.

Dependent Variable	Model	LOOIC/KFOLDIC	Dependent Variable 2	R-Squared EG	Total R-Squared
COP distance	null model	3106	EO	-	-
			EC		
	measurement	3062	EO	0.07 [0.04, 0.12]	0.64 [0.54, 0.71]
			EC	0.01 [0.00, 0.03]	0.51 [0.39, 0.61]
COP sway area	null model	3087	EO	-	-
			EC		
	measurement	3082	EO	0.01 [0.00, 0.02]	0.70 [0.60, 0.76]
			EC	0 [0.00, 0.01]	0.58 [0.46, 0.67]
COP velocity	null model	656	EO	-	-
			EC		
	measurement	647	EO	0.07 [0.02, 0.13]	0.30 [0.16, 0.42]
			EC	0.01 [0.00, 0.04]	0.23 [0.10, 0.36]

COP—center of pressure; LOOIC—leave-one-out information criterion; KFOLDIC—k-fold information criterion EO—eyes open; EC—eyes closed; R-squared EG stands for R-squared for the measurement effect. Total R-squared is a variance explained by a random effect to a study participant.

**Table 2 ijerph-20-02198-t002:** Fitness of the regression models to the dataset used in this paper for testing differences between groups.

Dependent Variable	Model	LOOIC/KFOLDIC	Dependent Variable 2	R-Squared EG	Total R-Squared
COP distance	null model	4130	EO	-	-
			EC		
	EG	4127	EO	0.03 [0.01, 0.07]	0.49 [0.36, 0.60]
			EC	0.02 [0.00, 0.07]	0.50 [0.39, 0.59]
	EGI	4129	EO	0.04 [0.01, 0.08]	0.52 [0.39, 0.62]
			EC	0.03 [0.00, 0.07]	0.51 [0.39, 0.60]
COP sway area	null model	4329	EO	-	-
			EC		
	EG	4329	EO	0.01 [0.00, 0.03]	0.32 [0.22, 0.47]
			EC	0.01 [0.00, 0.03]	0.34 [0.20, 0.5]
	EGI	4332	EO	0.01 [0.00, 0.03]	0.33 [0.22, 0.47]
			EC	0.01 [0.00, 0.03]	0.34 [0.20, 0.5]
COP velocity	null model	877	EO	-	-
			EC		
	EG	881	EO	0.02 [0.00, 0.05]	0.04 [0.00, 0.12]
			EC	0.01 [0.00, 0.04]	0.05 [0.00, 0.14]
	EGI	874	EO	0.06 [0.02, 0.13]	0.09 [0.03, 0.18]
			EC	0.02 [0.00, 0.05]	0.05 [0.01, 0.15]

COP—center of pressure; EG—main effects model; EGI—main effects and interaction model; LOOIC—leave-one-out information criterion; KFOLDIC—k-fold information criterion; EO—eyes open; EC—eyes closed; R-squared EG stands for R-squared for the measurement effects and a group. Total R-squared is a variance explained by a random effect to a study participant.

## Data Availability

Data sets generated and/or analyzed during the current study are available from the corresponding author on reasonable request.

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
