# Peer review of "Analysis of Postural Stability Following the Application of Myofascial Release Techniques for Low Back Pain—A Randomized-Controlled Trial"

_ijerph, 2023, doi:10.3390/ijerph20032198_

Round 1

Reviewer 1 Report

The authors present a RCT of myofascial release for CLBP. They randomize patients to an experimental group of MFR with posturography stabilometric assessment immediately after and one month after. They find very few differences between the two groups. 

Overall this is an interesting study that is generally well carried out. We do think there are some things that need attention to strengthen the manuscript. 

What are sway areas? The authors need to expand what MFR really is in the methods section. Why not include an assessment for the control group right after their PT session without MFR?

Extensive English editing is required:

            Abstract: “a one MFR” to “one MFR”

            Intro: “when lasting over 3 months or persistent even after healing damaged tissues is especially problematic, because CLBP occurrence increases with age, reaching its peak in the 5th and 6th decades of life”. Run on, disconnect sentences. 

            ‘whereas even 90%” to “whereas 90%”

The authors conclude that the reason no differences were seen is that there weren’t enough MFR treatments and not enough body area involved. Could anything else be the case? Perhaps the literature for MFR needs re-examination? Could it be that it really is not effective?

Reviewer 2 Report

The study did not specify the average degree of pain intensity in each group. The number of women and men in the groups was as well not specified, which is important because both sexes react differently to pain and treatment, especially in terms of the emotional factor. Groups can be heterogeneous.

Without additional measuring devices, it is impossible to determine whether the type and strength of the MFR intervention were identical in each patient. It is known that muscle tension depends on the intensity of pain and vice versa, as in the vicious circle of pain. Therefore, it would also be important to examine the muscle tone of the subjects.

Is there evidence that the fascia in LBP is a source of pain in examinated groups? Were imaging tests performed to exclude the possible cause of pain in the intervertebral discs (the disc itself, without compression of the nerve structures, can be a source of pain) and adjacent structures, e.g. anomalies in the structure of the spine? An attempt to objectify the MFR by means of stabilometry is valuable but insufficient. It would be advisable to study using a measuring tool for the applied intervention, as well as to study the correlation of stabilometry with pain intensity and muscle tension.
